# The Effect of Smoking during Pregnancy on Severity and Directionality of Externalizing and Internalizing Symptoms: A Genetically Informed Approach

**DOI:** 10.3390/ijerph17217921

**Published:** 2020-10-28

**Authors:** Mikael O. Ekblad, Kristine Marceau, Emily Rolan, Rohan H. C. Palmer, Alexandre Todorov, Andrew C. Heath, Valerie S. Knopik

**Affiliations:** 1Department of General Practice, Institute of Medicine, Turku University and Turku University Hospital, 20014 Turku, Finland; 2Department of Human Development and Family Studies, Purdue University, West Lafayette, IN 47906, USA; kristinemarceau@purdue.edu (K.M.); vknopik@purdue.edu (V.S.K.); 3Department of Psychology, Michigan State University, East Lansing, MI 48824, USA; doneremi@msu.edu; 4Department of Psychology, Behavioral Genetics of Addiction Laboratory, Emory University, Atlanta, GA 30322, USA; rohan.palmer@emory.edu; 5Department of Psychiatry, Midwest Alcoholism Research Center, Washington University School of Medicine, Saint Louis, MO 63110, USA; todorov@wustl.edu (A.T.); aheath@wustl.edu (A.C.H.)

**Keywords:** genetically-informed designs, prenatal exposure, tobacco, health consequences, sibling comparison

## Abstract

The objective was to examine the association between maternal smoking during pregnancy (SDP) and (I) severity and (II) directionality of externalizing and internalizing symptoms in a sample of sibling pairs while rigorously controlling for familial confounds. The Missouri Mothers and Their Children Study is a family study (N = 173 families) with sibling pairs (aged 7 to 16 years) who are discordant for exposure to SDP. This sibling comparison study is designed to disentangle the effects of SDP from familial confounds. An SDP severity score was created for each child using a combination of SDP indicators (timing, duration, and amount). Principal component analysis of externalizing and internalizing behavior, assessed with the Child Behavior Checklist and Teacher Report Form, was used to create symptom severity and directionality scores. The variance in severity and directionality scores was primarily a function of differences between siblings (71% and 85%, respectively) rather than differences across families (29% and 15%, respectively). The severity score that combines externalizing and internalizing symptom severity was not associated with SDP. However, a significant within-family effect of SDP on symptom directionality (b = 0.07, *p* = 0.04) was observed in the sibling comparison model. The positive directionality score indicates that SDP is associated with differentiation of symptoms towards externalizing rather than internalizing symptoms after controlling for familial confounds with a sibling comparison model. This supports a potentially causal relationship between SDP and externalizing behavior.

## 1. Introduction

Maternal smoking during pregnancy (SDP) remains one of the most common environmental exposures compromising fetal health. Despite a decreasing trend of SDP in most developed countries, research suggests that young and low-educated women continue to be at a higher risk for SDP and over 50% of women who smoke during early pregnancy continue to smoke throughout pregnancy [1]. Critically, SDP is associated with multiple adverse pregnancy complications [2] as well as prenatal and perinatal outcomes, including preterm birth, low birth weight, smaller head circumference, and an increased risk for sudden unexpected infant death [3,4]. Further, exposure to SDP seems to alter child brain structure and function [5,6], which may be the starting point for a cascade of alterations in offspring behavior across the life span.

Multiple studies have linked SDP and externalizing problems across diverse samples and developmental periods [7,8,9,10,11,12]. Alternatively, few studies have found an association between SDP and internalizing problems [13,14] with the majority of studies reporting null results [8,10,15]. Externalizing behaviors are characterized by rule breaking, hyperactive, and aggressive behavior. These may predispose later juvenile delinquency, adult crime, and violence [16]. Additionally, internalizing behavior are characterized by withdrawn, anxious, and depressed behavior that is not always evident to others [16]. Early internalizing behaviors have been linked to later depressive and mental health symptoms and lower educational attainment [17].

It has become essential for studies examining the effect of SDP on child behavior to use quasi-experimental designs, for example comparing siblings with discordant exposure to SDP, to disentangle the effects of SDP from familial confounding factors, including genetic influences [4,18]. Studies using genetically informed designs have provided evidence that genetic and family factors explain a portion, or all, of the association between SDP and externalizing problems [12,19,20,21,22]. In regard to internalizing behavior, to our knowledge, there has been only one study using a genetically informed study that found no association between SDP and internalizing behavior [15].

It is also important to incorporate reports from multiple informants that view the child from different perspectives and in different contexts, to be able to gain a more comprehensive picture of child behavior [23,24,25]. For example, we found in our previous sibling design study [12] a significant association between SDP and disruptive behavior when using a multi-rater approach that capitalizes on both parent and teacher report. On the contrary, two previous studies found that the effect of SDP on externalizing behavior, assessed only by maternal report, was fully attenuated utilizing a sibling comparison design [19,20].

Both externalizing and internalizing behavior, as outcomes and predictors of later behavior, have been studied extensively in youth [26]. The main methodological issue of previous studies stems from a failure to account for (I) the epidemiological comorbidity of externalizing and internalizing behavior, as well as, (II) that the risk factors for both appear much the same [24,26,27,28]. In most studies, the effect of SDP on these behaviors has usually been studied separately because of the collinearity problem if both measures are included simultaneously into the model [27]. A handful of studies on externalizing and internalizing behavior have addressed this issue by creating (I) a severity score of the two domains that represents what the two domains have in common, and (II) a directionality score that represents how the domains differentiate, that is, whether the child has a propensity towards externalizing vs. internalizing problems regardless of symptom severity [24,27,28,29]. These scores are orthogonal and can be analyzed separately. We use this unique method to account for co-morbidity of externalizing and internalizing behavior within individuals.

The present study sought to examine the associations between SDP and (I) severity and (II) directionality of externalizing and internalizing behavior assessed by multiple reporters. Further, we use a sibling design approach that can be used to strengthen causal inferences regarding environmental risks. This design controls rigorously for familial confounding including genetic influences.

## 2. Methods

The Missouri Mothers and Their Children study (MO-MATCH) is a family study specifically designed to leverage the sibling comparison approach to examine the effects of prenatal smoking exposure on child behavioral problems and associated learning and cognitive deficits [30]. Families of the MO-MATCH study were identified using birth records obtained from the Missouri Department of Health and Senior Services Bureau of Health Informatics. Families where mothers changed smoking behavior between two pregnancies for child’s birth years 1998–2005 were identified (N over 4000). A screening interview was conducted for 1520 families to verify SDP information from the birth record. The mothers (N = 413) whose interview information (I) confirmed a history of regular smoking (i.e., smoking more than 100 cigarettes lifetime) [31] and (II) aligned with the birth record, were invited to participate in the study. Families were excluded if (1) mothers failed to understand the elements of informed consent; (2) English was not the primary language spoken in the home; (3) children had a history of head trauma, neurological disorders, or uncorrected visual or auditory acuity deficits; and (4) mothers used nicotine substitutes in the “nonsmoking” pregnancy. The final study population consisted of 173 families, including 344 pregnancies (including both siblings from each family). A total of 54% (N = 94) of fathers participated in the study. Ninety-six percent (N = 250) of the parents were of Caucasian ancestry. In general, the MO-MATCH sample is comparable to the state of Missouri in marriage rates and rates of parents reporting completing “some college;” however, MO-MATCH shows higher rates of tobacco use than state averages because of the recruiting criteria. The sample demographics are presented in Table 1.

The study was approved by the Institutional Review Boards of Rhode Island Hospital (IRB 1—#00000396 and IRB 2—#00004624), Washington University (#FWA00002284), the State of Missouri Department of Health and Senior Services (#10-0300), and Purdue University (#FWA00001548).

### 2.1. Smoking during Pregnancy

A modified version of the Missouri Assessment of Genetics Interview for Children (MAGIC)—Parent on Child was used to obtain information on maternal self-report of SDP [32]. The MAGIC interview has a high reliability (kappa = 0.60−0.66) and stability (kappa = 0.95) for maternal reporting about their pregnancies including information on smoking behavior [33]. Previously, retrospective self-report of maternal SDP, compared to the predictive utility of both birth record report and paternal report of maternal SDP, was found to be the best measure of SDP in the MO-MATCH data [34]. Thus, the following indicators were used to create a SDP severity score: (a) Any SDP (No = 0, Yes = 1) across each pregnancy as a whole and specific to each trimester, and (b) overall quantity smoked during pregnancy via mothers’ estimate of the number of cigarettes smoked in each trimester. Continued SDP has been found to be potentially more harmful [35,36], and thus, the SDP severity score included also timing of exposure. A single SDP severity score was created for each child consistently with prior reports [12,21,22,34,37]:did not smoke during pregnancy;smoked during 1st trimester only, cigarettes: 1–10 per day;smoked during 1st trimester only, cigarettes: 11–19 per day;smoked during 1st trimester only, cigarettes: 20+ per day;smoked beyond 1st trimester, cigarettes: 1–10 per day (maximum of all trimesters);smoked beyond 1st trimester, cigarettes: 11–19 per day (maximum of all trimesters); andsmoked beyond 1st trimester, cigarettes: 20+ per day (maximum of all trimesters).

The sensitivity analyses included the following alternative SDP variables: (I) SDP yes/no: a binary indicator of whether mothers smoked during a given pregnancy, (II) Sum quantity across trimesters: a continuous sum score of the quantity variables for each trimester, (III) Maximum quantity: a continuous variable which was equal to the quantity for the single trimester in which the mother reported smoking the most, and (IV) Number of trimesters smoked: an ordinal variable indicating the number of trimesters in which mother reported SDP (0, 1, 2, or 3). The correlations between the single SDP severity score and each of these alternative SDP variables were as follows: SDP severity with SDP yes/no (r = 0.80, *p* < 0.05), with Sum quantity (r = 0.74, *p* < 0.05), with Max quantity (r = 0.69, *p* < 0.05), and with N trimesters smoked (r = 0.94, *p* < 0.05).

### 2.2. Child Behavior

Child externalizing and internalizing behavior was assessed via maternal-report on the Child Behavior Checklist (CBCL) [38] and teacher-report on the Teacher Report Form (TRF) [39]. The coefficient alpha’s for internalizing was 0.90 and for externalizing symptoms was 0.94 for the CBCL and 0.90 and 0.95 for the TRF, respectively [40]. The test-retest reliabilities were high for both internalizing (0.91) and externalizing symptoms (0.92) for the CBCL, as well as, for the TRF (0.86 and 0.89, respectively) [40]. The children were 7–16 years old when the assessments occurred. The mean age difference between the siblings was 2.8 years (SD = 1.54) (Table 1). The CBCL and TRF assess behavioral symptoms over the previous 6 months on a scale of 0 (not true) to 2 (very true/often true). The TRF was available for only part of the study population (220 of 344 children). TRF data was missing at random in relation to SDP severity (χ^2^, *p* = 0.23) and birth order (χ^2^, *p* = 0.71). We had reports from two different teachers in 95 cases, and thus similarly to prior reports [12,21], we used the “or” rule to obtain a single, maximum teacher-rated score per child for each item in these cases [41,42]. Having only one vs. two TRF was random in relation to SDP severity (χ^2^, *p* = 0.62) and birth order (χ^2^, *p* = 0.27).

The multi-rater scores were created identical to our prior reports for whom the data was available [12,21]. The “or” rule was applied for each item that was assessed both in CBCL and TRF [41,42]. If the item was asked only on the CBCL or on the TRF, this was included directly in the maximized multi-rater score. The externalizing multi-rater score consisted of 39 items (Cronbach’s α = 0.92) and the internalizing multi-rater score of 35 items (α = 0.88).

To include both externalizing and internalizing symptoms in the same model without issues of multicollinearity (correlation of 0.49, *p* < 0.001), we conducted a principal component analysis of the externalizing and internalizing multi-rater scores to create symptom severity and directionality scores [27,29]. The severity score represents what the two scores have in common, i.e., symptom severity, regardless of whether the symptoms fall on the internalizing versus externalizing spectrum. A high severity score indicates high levels of problems, whereas a low severity score indicates low levels of problems regardless of type. The externalizing and internalizing symptoms loaded equally and strongly onto the severity score (factor loadings = 0.86). The symptom directionality score represents what differentiates the two behavior scores. The factor loadings for the directionality score are equal in weight but opposite in direction for externalizing (0.51) and internalizing (−0.51) symptoms. This means that a high positive directionality score indicates a propensity toward externalizing symptoms and high negative directionality score indicates a propensity toward internalizing symptoms. Together, severity and directionality scores account for 100% of the variance in symptoms. The severity score explained 74% of the variance.

We performed sensitivity analyses by using the separate externalizing and internalizing multi-rater scores instead of severity and directionality scores. The residuals of the hypothesis-testing models were skewed, thus, the multi-rater composite scores were square root transformed.

### 2.3. Child and Familial Covariates

We chose to use consistent covariates with previous genetically informed studies of SDP and behavioral outcomes [12,19,21,22,43]. These comprised of (1) maternal age, marital status, and education at birth of each child; (2) child sex and birth order; and (3) second-hand smoke exposure during pregnancy by the father and qualification for food stamps of the families at the time of delivery. The information on families’ qualification for food stamps was used to control for other maternal and family characteristics that could potentially confound the association of SDP and behavioral symptoms. Birth order correlated significantly and negatively with age (r = −0.87) leading to a multicollinearity problem when modeling these data. Consistent with previous studies [12,21,22], birth order was included as a covariate rather than age because most of the mothers (64%) smoked during the first pregnancy but not during the second.

### 2.4. Statistical Analysis

In this sibling comparison study, we used SAS PROC MIXED (SAS 9.4 System. Cary, NC, USA, SAS Institute Inc) in fitting a series of hierarchical linear models to account for non-independence of data. Hierarchical linear models allow for examination of both the within- and between-family associations of SDP with child behavior. Our approach was identical to previous MO-MATCH studies that have similarly investigated the association between SDP and child behavioral outcomes [12,21,22,37].

First, an unconditional “intercept-only” model was fit to the data to decompose the variance in the behavior into within-family (e.g., individual child-level) and between-family (e.g., family level) variation via intra-class correlations (ICC) [44]. ICC for between-family variation was calculated as the (family-level variance (family-level variance + individual child-level variance)). The unconditional model provides a baseline against which subsequent models are compared in order to understand how much within-family (and therefore potentially causal) variance SDP and covariates explain separately in each outcome measure.

Second, we fit a standard model to the data (Model 1). This model was used as an approximation of the standard comparison seen in the literature, which does not capitalize on the sibling comparison aspect of the data or the family structure; although, this model does adjust for the nesting structure of children within families. The covariates listed above were included in this model.

Third, the sibling comparison model was fit to the data (Model 2). This evaluates siblings, that are discordant for exposure to SDP, within families (i.e., mother smoked [or smoked more] while pregnant with one sibling and did not smoke (or smoked less) while pregnant with the other sibling). The nature of the design matches the siblings for confounding familial factors. Model 2 examined whether SDP is operating at the within-family level (e.g., contributing to differences in behavior in one sibling vs. another, within families) or between-family level (e.g., contributing to differences in overall, average levels of siblings’ behavior across families).

For Model 2, a “family average SDP severity score” and a “child-specific SDP severity score” were created to specifically assess both within- and between-family associations of SDP and outcome measures [31]. The “child-specific SDP severity” assessed the within-family effect of SDP on outcome measures, which is a test of any unique effect of SDP on child specific outcomes over and above familial and genetic factors that siblings share. The average score for SDP severity across both siblings was used to obtain a “family average SDP severity score” for each family, which was used to assess the between-family effect of SDP on outcome measures. “Child-specific SDP severity” for each child was the resulting value when the “family average SDP severity score” was subtracted from each child’s “SDP severity score” used in Model 1 (i.e., within-family centering). Thus, the sibling for whom mothers smoked (or smoked more) would have a positive score of “child-specific SDP severity,” whereas the sibling for whom mothers did not smoke (or smoked less) would have a negative score of the same magnitude.

The detailed hierarchical linear model equation used for Model 2 can be found in Knopik et al. [21]. In Model 2, the “child-specific SDP severity” score was entered as a level 1 predictor, and the family average SDP severity score was entered as a level 2 predictor. The covariates were included into the Model 2 in the following way: the covariates that differed for siblings 1 and 2 (mother age at childbirth, mothers’ education at childbirth, child sex, secondhand smoke exposure, food stamp use at childbirth) were separated into child-specific scores relative to family average and family average components, similar to SDP as described above [12,21,22,37]. Thus, we controlled for both the within- and between-family effects of covariates (with separate child-specific variables). Within-family covariates were also centered as such. Marital status did not differ for any siblings within a family, and thus was not separated into within- and between-family components.

In sum, we fit one unconditional model and two conditional models (Models 1 and 2) for each outcome variable. In order to quantify how much of the within-family variance is explained by each conditional model, we computed the percentage of the (within-family) variance explained: ((unconditional individual child-level variance—conditional, e.g., Model 1, individual child-level variance)/unconditional individual child-level variance) [45].

## 3. Results

The findings between SDP-symptom severity and directionality scores, including more specific information e.g., on between-family effects and percentage of the within-family variance explained by each conditional model, are summarized in Table 2. The results of sensitivity analyses of SDP and the separate externalizing and internalizing multi-rater scores are presented in Appendix A. Consistent with our previous works [12,21,22,37], the results of the sensitivity analyses that incorporate multiple alternative measurements of SDP are presented in Appendix A. Child sex was generally a significant predictor throughout the models.

### 3.1. Symptom Severity

As noted above, the severity score represents symptom severity, regardless of whether the symptoms fall on the internalizing versus externalizing spectrum. A high severity score indicates high levels of problems, whereas a low severity score indicates low levels of problems regardless of type. The variance in the unconditional model for symptoms severity was primarily a function of differences between siblings (within-family variability 71%) rather than differences across families (between-family differences 29%). SDP and symptom severity was not significantly associated in the standard (Model 1; b = 0.04, *p* = 0.13) nor in the sibling comparison model (Model 2; b = 0.03, *p* = 0.40).

### 3.2. Symptom Directionality

As previously discussed, the directionality score represents what differentiates externalizing from internalizing scores. High positive directionality scores indicate a propensity toward externalizing symptoms and high negative directionality scores indicate a propensity toward internalizing symptoms. The variance in symptom directionality was primarily a function of within-family variability 85% (differences between siblings) rather than between-family differences 15% (difference across families). A trend-level association between SDP and directionality (b = 0.06, *p* = 0.06) was found in the standard model (Model 1). However, a significant within-family effect of SDP on symptom directionality (b = 0.07, *p* = 0.04) was shown in the sibling comparison model (Model 2). The positive association with the directionality score indicates that SDP was associated with symptom differentiation toward externalizing symptoms compared to internalizing symptoms.

### 3.3. Sensitivity Analyses

The sensitivity analyses were performed by using the separate externalizing and internalizing multi-rater scores instead of severity and directionality scores. As expected, SDP predicted greater externalizing behavior in the standard model (Model 1; b = 0.10, *p* = 0.01) as well as in the sibling comparison model (Model 2; b = 0.09, *p* = 0.03). SDP severity was not associated with internalizing behavior in the standard (Model 1) nor the sibling comparison model (Model 2) (Appendix A). In sum, the sensitivity analyses that considered the effect of different definitions of SDP on the outcome measures showed a similar pattern of results (Appendix A).

## 4. Discussion

In this genetically informed study that included sibling pairs discordant for exposure to SDP, we were able to examine the association between SDP and (I) severity and (II) directionality of externalizing and internalizing behavior assessed using multiple raters (i.e., parents and teachers). SDP was associated with positive directionality scores, meaning that SDP predicted differentiation of behavior problems toward externalizing behavior. However, there was no effect of SDP on the severity score, meaning that SDP did not associate with combined symptom severity of externalizing and internalizing behavior. This pattern of results remained in sensitivity analysis. Overall, the current study using a genetically informed approach provides additional evidence that SDP is linked with later externalizing behavior even after accounting for the epidemiological comorbidity of externalizing and internalizing behavior.

A unique aspect of this study compared to previous studies was the use of severity and directionality scores to be able to account for co-morbidity of externalizing and internalizing behavior that occurs within-individuals. The main findings of the effect of SDP on severity and directionality are rational and are in line with the findings from previous studies [7,8,9,10,11,12,13,14,15] and the sensitivity analyses of this current study with more traditional externalizing and internalizing outcome measures, i.e., SDP has been associated with externalizing behavior but not with internalizing behavior. That SDP predicted differentiation towards externalizing symptoms is consistent with the generalist gene and specialist environments hypothesis, which suggests that genetic influences should be more predictive of severity of problems and comorbidity, whereas environmental influences should explain their divergence [28,46,47]. Here, we find support that SDP may be a specific environmental influence serving to nudge individuals down developmental trajectories that are more externalizing in nature, perhaps through impairments in brain regions that are linked to behavioral disinhibition (a hallmark of externalizing but not internalizing problems) [48]. Interestingly, the only other predictor of directionality of internalizing/externalizing symptoms in the literature was also a prenatal influence: obstetric complications predicted a preponderance of internalizing relative to externalizing problems.

In the context of genetic risk and marital hostility [29], our findings contribute support for an emerging hypothesis that prenatal exposures may be particularly important for understanding the directionality of symptoms in childhood and adolescence [28]. The lack of finding for severity is consistent with the idea that SDP affects underlying neural structures as described above, but that it takes some other developmental influence (e.g., generalist genes) to push youth towards exhibiting more severe symptoms.

A major strength of this study is the use of a family study, specifically designed to examine the effects of SDP on child behavior and to control for familial confounds, including genetic influences. Thus, our findings bolster the expanding literature showing the need for genetically informed studies, as well as studies that account for behavioral co-morbidity to disentangle the true effect of SDP on later behavior.

We note the following limitations that should be considered when interpreting the results of the current study. First, the information on SDP was based on maternal self-report, which generally underestimates the true prevalence of SDP [49]. Further, there may be a risk for recall bias when assessing specific details about the quantity of SDP [50]. However, the retrospective report of SDP has been shown to be reliable in this particular study population [34]. In addition, our results were consistent in the sensitivity analyses that utilized varying SDP measures that differentially considered amount, timing, and duration of SDP.

Second, parents reported on both SDP and child behavior, which may introduce measurement bias. This potential bias further highlights the need for including both parent and teacher reports of behavior. Third, we are aware that while the sensitivity analyses were consistent with our hypothesized findings, our results do not survive a conservative Bonferroni correction. Finally, we chose the covariates based on previous genetically informed studies examining externalizing and internalizing behavior, but there are undoubtedly other factors that differ between siblings and hence may influence the sibling comparison [19].

## 5. Conclusions

In conclusion, the significant finding between SDP and a positive symptom directionality score, interpreted as a symptom differentiation toward externalizing symptoms, supports a potentially causal relationship between SDP and externalizing behavior. However, this finding should be interpreted cautiously given the study limitations. Nonetheless, this study suggests that it is too early to state that there is no association between SDP and later externalizing behavior, particularly when considered the negative effect of SDP on fetal brain development. Consistent with the few studies to date, our results do not support an association between SDP and internalizing behavior.

## Figures and Tables

**Table 1 ijerph-17-07921-t001:** Sample characteristics.

Study Variables	Child 1			Child 2		
	N	Mean	SD	N	Mean	SD
SDP Severity	173	3.95	2.05	171	2.04	1.77
Outcome variables						
Externalizing symptoms ^1^	165	9.20	8.00	164	9.63	9.60
Internalizing symptoms ^1^	165	9.68	8.35	164	8.93	7.33
Severity	165	0.01	1.00	164	−0.01	1.01
Directionality	165	−0.07	0.96	164	0.07	1.04
Covariates						
Maternal age at birth	162	26.48	5.55	163	29.22	5.75
Maternal education (in years) at birth	162	13.28	2.12	163	13.50	1.94
Second-hand smoke exposure by fathers	171	1.84	1.44	161	1.15	1.43
	**N**	**%**		**N**	**%**	
Sex (male)	91	53		87	51	
Marital status (percent married) at birth	155	85		159	83	
Food stamp usage at birth	149	9		150	13	
Family demographics (at assessment)	N					
Maternal age	162	39.83				
Paternal age	80	44.04				
Child 1 age	173	12.99				
Child 2 age	170	10.19				
Child age difference	170	2.79				
	**Maternal**			**Paternal**		
	**N**	**%**		**N**	**%**	
Education						
Less than High school	7	4		9	10	
High school	30	18		19	20	
1–2 years college	50	30		14	15	
3–4 years college	46	27		17	18	
More than college	29	17		21	22	
Not reported	7	4		14	15	
Mother’s marital status						
Never married	6	4				
Married	130	77				
Separated	5	3				
Divorced	26	15				
Widowed	2	1				

Smoking during pregnancy (SDP). N reflects the sample size with valid data and vary due to random missing data in the birth record. ^1^ The externalizing and internalizing scores are presented as raw scores, which were square root transformed due to skew in the residuals for all the analyses.

**Table 2 ijerph-17-07921-t002:** Model Results of SDP Severity and Directionality Associations.

	Severity						Directionality					
	Unconditional Model		Model 1— Standard		Model 2— Sib-Comparison		Unconditional Model		Model 1— Standard		Model 2— Sib-Comparison	
	*b*	SE	*b*	SE	*b*	SE	*b*	SE	*b*	SE	*b*	SE
Intercept	<0.01	0.06	<−0.01	0.48	0.26	0.48	<0.01	0.06	−0.02	0.46	0.88	0.56
SDP												
Child-specific SDP			0.04	0.03	0.03	0.03			0.06 †	0.03	0.07 *	0.03
Family average SDP					0.13 *	0.06					0.05	0.06
Controls												
Child-specific sex			−0.28 *	0.1	−0.27 *	0.12			−0.23 *	0.10	−0.18	0.13
Family average sex					−0.29	0.18					−0.35 †	0.18
Birth order			0.04	0.12	0.04	0.21			0.38 *	0.10	0.23	0.20
Child-specific mother education			<0.01	0.03	0.08	0.06			<0.01	0.03	0.07	0.06
Family average mother education					−0.03	0.04					−0.02	0.04
Child-specific mother age			<−0.01	0.01	−0.02	0.05			−0.01	0.03	0.04	0.06
Family average mother age					<0.01	0.01					−0.01	0.01
Mother marital status			0.08	0.18	0.07	0.18			−0.12	0.19	−0.10	0.20
Child-specific food stamps			0.24	0.18	0.45 *	0.22			−0.06	0.19	−0.08	0.27
Family average food stamps					0.01	0.21					−0.09	0.27
Child-specific Second-hand smoke exposure			0.07	0.04	0.09	0.08			<0.01	0.04	−0.03	0.07
Family average Second-hand smoke exposure					0.04	0.05					0.03	0.05
Variance												
Family-level	0.29 *		0.38		0.41 *		0.15 *		0.70 *		0.39 *	
Random-effect on SDP			0.02		0.02				0.05 *		0.04 *	
Individual-level (residual)	0.71 *		0.44 *		0.42 *		0.85 *		0.30 *		0.34 *	
% within-family variance explained beyond unconditional model			38.22%		41.29%				64.59%		60.55%	
Model Fit												
−2 Res Ln L	923.1		772.1		782.4		933.1		759.7		775.9	
AIC	927.1		780.1		790.4		937.1		767.7		783.9	

* *p* < 0.05, † *p* < 0.10. In the Sibling-comparison model, parameters are “child-specific” relative to family average.

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
