# Peer review of "The Effect of Smoking during Pregnancy on Severity and Directionality of Externalizing and Internalizing Symptoms: A Genetically Informed Approach"

_ijerph, 2020, doi:10.3390/ijerph17217921_

Round 1

Reviewer 1 Report

Dear authors,

I have reviewed the manuscript entitled "The Effect of Smoking During Pregnancy on Severity and Directionality of Externalizing and Internalizing Symptoms: A Genetically-Informed Approach". It is a carefully  designed study and a well written manuscript.

Minor comments to address:  

Consider to include a definition for :

  • externalizing/internalizing behavior and respective symptoms
  • directionality

 L. 110 should read :Thus, the following indicators were used to create a SDP severity score

L. 302-303 should read :Third, we are aware that while the sensitivity analyses were consistent with our  hypothesized findings, our results do not survive a conservative Bonferroni correction

Results:

  • supplementary tables are differently labeled in the manuscript text and in the supplementary material where they appear as supplementary table 3 and 4,  and  supplementary table 1 and 2 respectively.
  •  supplementary tables could be included in the results section of main manuscript body unless it exceeds nr of tables permitted 

Author Response

Minor comments to address:  

Consider to include a definition for :

  • externalizing/internalizing behavior and respective symptoms
  • directionality

Thank you for this comment. We have added the definition for externalizing and internalizing symptoms to the introduction (rows 60-64). The definition for directionality is mentioned on rows 86-89.

  1. 110 should read :Thus, the following indicators were used to create a SDP severity score

We have edited this sentence (row 152 in the reviewed version of the manuscript).

  1. 302-303 should read :Third, we are aware that while the sensitivity analyses were consistent with our  hypothesized findings, our results do not survive a conservative Bonferroni correction

We have edited this sentence (row 401-402 in the reviewed version of the manuscript)

Results:

  • supplementary tables are differently labeled in the manuscript text and in the supplementary material where they appear as supplementary table 3 and 4,  and  supplementary table 1 and 2 respectively.

Thank you for noticing this error. We have relabeled the supplementary tables to the main text (rows 307-308 and 345-346).

  • supplementary tables could be included in the results section of main manuscript body unless it exceeds nr of tables permitted 

Thank you for suggesting the addition of supplementary tables to the main manuscript. We prefer keeping the supplementary tables separate from the main text in order to highlight the main results of this study.  The supplementary tables include results from the sensitivity analyses. However, if the editors would prefer otherwise, we will add the supplementary tables to the main text.

Reviewer 2 Report

The study aimed to examine the association between maternal smoking during pregnancy (SDP) and (i) severity and (ii) directionality of externalizing and internalizing symptoms in a sample of sibling pairs while controlling for familial confounds. The Missouri Mothers and Their Children Study is a family study (N=173 families) with sibling pairs (aged 7 to 16 years) who are discordant for exposure to SDP. This sibling comparison study is designed to disentangle the effects of SDP from familial confounds. An SDP severity score was created for each child using a combination of SDP indicators (timing, duration, and amount). Principal component analysis of externalizing and internalizing behavior, assessed with the Child Behavior Checklist and Teacher Report Form, was used to create symptom severity and directionality scores. The authors found that the variance in severity and directionality scores was primarily a function of differences between siblings (71% and 85%, respectively) rather than differences across families (29% and 15%, respectively). The severity score that combines externalizing and internalizing symptom severity was not associated with SDP. However, a significant within-family effect of SDP on symptom directionality (b=0.07, p=0.04) was observed in the sibling comparison model. Therefore the authors concluded that the positive directionality score indicates that SDP is associated with differentiation of symptoms towards externalizing rather than internalizing symptoms after controlling for familial confounds with a sibling comparison model. This supports a potentially causal relationship between SDP and externalizing behavior. However, there were some points that could be clarified for a better understanding of the readers.

  1. There were no explanations for the terms “externalizing symptoms” and “internalizing symptoms”. Please clarify.
  2. The scale and questionnaire used in the study, such as MO-MATCH, MAGIC, CBCL, and TRF, should be clarified with a description of their validity and reliability reported in previous research.
  3. The “genetically-informed approach” was mentioned in the abstract and also the main text. However, it is hard to find the relationship between SDP and genetic action from the message available in the manuscript. Please clarify.

Author Response

1. There were no explanations for the terms “externalizing symptoms” and “internalizing symptoms”. Please clarify.

Thank you for this comment.  Reviewer 1 requested the same. We have added the definition for externalizing and internalizing symptoms to the introduction (rows 60-64).

2. The scale and questionnaire used in the study, such as MO-MATCH, MAGIC, CBCL, and TRF, should be clarified with a description of their validity and reliability reported in previous research.

Thank you for this comment. The study design article of this study by Knopik et al. 2015 includes information on these. As suggested, we have also added this information to the methods section (on rows 147-149 and 176-180).

3. The “genetically-informed approach” was mentioned in the abstract and also the main text. However, it is hard to find the relationship between SDP and genetic action from the message available in the manuscript. Please clarify.

Thank you for this comment. In this study, we have used a sibling comparison design, which is one of several quasi-experimental designs that can be used to strengthen causal inferences regarding environmental risks by ruling out specific forms of confounding, including confounding by gene-environment correlation. Please see rows 65-71 and 92-94 of the introduction.

Reviewer 3 Report

I am grateful for the opportunity to review the manuscript presented to me. I hope that the comments in the review would be helpful
 to publish the manuscript. I believe the paper is worth considering for publication, however requires minor revision.

  1. In the introductory part, please clearly emphasize the uniqueness of the presented research
  2. Methods:"Families were excluded if... "English was not the primary language spoken in the home" - what was the reason?
  3. in my opinion, the discussion part should be expanded: it is relatively short for introduction and methodology.
  4. the conclusion part is too extensive - please summarize it
  5. other minor remarks in the attached text

Author Response

I am grateful for the opportunity to review the manuscript presented to me. I hope that the comments in the review would be helpful to publish the manuscript. I believe the paper is worth considering for publication, however requires minor revision.

We appreciate the kind words of our work.

1. In the introductory part, please clearly emphasize the uniqueness of the presented research

Thank you for this comment. We have added a sentence of the uniqueness of this study to the introduction as suggested in this comment and in the separately attached document.

2. Methods:"Families were excluded if... "English was not the primary language spoken in the home" - what was the reason?

Thank you for this comment. In this study, questionnaires only in the English language were used and  maternal interviews were conducted in English. Including families with possibly weak English language skill could have affected the reliability of the results. Thus, only families with English as the primary language spoken in the home were included.

3. in my opinion, the discussion part should be expanded: it is relatively short for introduction and methodology.

4. the conclusion part is too extensive - please summarize it

Thank you for the comments 3 and 4. We have made minor modifications to the discussion and shortened the conclusion.

5. other minor remarks in the attached text

Thank you for noticing the small errors in the text. Please see the editions on rows 82 and 140.

Round 2

Reviewer 2 Report

The response and revised manuscript have addressed my concerns.